# Synergy Assessment of Four Antimicrobial Bioactive Compounds for the Combinational Treatment of Bacterial Pathogens

**DOI:** 10.3390/biomedicines11082216

**Published:** 2023-08-07

**Authors:** Kevin Masterson, Ian Major, Mark Lynch, Neil Rowan

**Affiliations:** 1Bioscience Research Institute, Technological University of the Shannon, N37 HD68 Athlone, Ireland; marklynch@tus.ie (M.L.); nrowan@tus.ie (N.R.); 2PRISM Research Institute, Technological University of the Shannon, N37 HD68 Athlone, Ireland; imajor@tus.ie

**Keywords:** antimicrobial resistance, bioactives, synergy analysis, drug combinations, two-drug combination, three-drug combination, four-drug combination, checkerboard assay, broth microdilution

## Abstract

Antimicrobial resistance (AMR) has become a topic of great concern in recent years, with much effort being committed to developing alternative treatments for resistant bacterial pathogens. Drug combinational therapies have been a major area of research for several years, with modern iterations using combining well-established antibiotics and other antimicrobials with the aim of discovering complementary mechanisms. Previously, we characterised four GRAS antimicrobials that can withstand thermal polymer extrusion processes for novel medical device-based and therapeutic applications. In the present study, four antimicrobial bioactive—silver nitrate, nisin, chitosan and zinc oxide—were assessed for their potential combined use as an alternative synergistic treatment for AMR bacteria via a broth microdilution assay based on a checkerboard format. The bioactives were tested in arrangements of two-, three- and four-drug combinations, and their interactions were determined and expressed in terms of a synergy score. Results have revealed interesting interactions based on treatments against recognised test bacterial strains that cause human and animal infections, namely *E. coli*, *S. aureus* and *S. epidermidis*. Silver nitrate was seen to greatly enhance the efficacy of its paired treatment. Combinations with nisin, which is a lantibiotic, exhibited the most interesting results, as nisin has no effect against Gram-negative bacteria when used alone; however, it demonstrated antimicrobial effects when combined with silver nitrate or chitosan. This study constitutes the first study to both report on practical three- and four-drug combinational assays and utilise these methods for the assessment of established and emerging antimicrobials. The novel methods and results presented in this study show the potential to explore previously unknown drug combination compatibility measures in an ease-of-use- and high-throughput-based format, which can greatly help future research that aims to identify appropriate alternative treatments for AMR, including the screening of potential new bioactives biorefined from various sources.

## 1. Introduction

Antimicrobial resistance (AMR) has become a topic of academic interest, as it has reached a crisis point that has driven scientists to consider novel appropriate solutions to overcome it [1,2,3]. An ever-increasing number of antibiotic resistant bacterial species have emerged that pose serious threats to modern medicine, causing a loss in efficacy of critical front-line therapeutics [4]. Antibiotics remain our primary means of eliminating pathological bacterial infections, and while there has been a recent resurgence in the development of novel antibiotic compounds, additional ways of tackling AMR bacteria are urgently needed [5,6]. Research that aims to discover alternative antimicrobials has been a major topic of interest, as it particularly hopes to circumvent the emerging resistance to mainstay antibiotics [7,8,9]. Additionally, the co-development of methods that can assess the efficacy of appropriate combinations of already established antimicrobial compounds is important to reduce reliance on a single treatment [10,11,12]. While a vast number of antimicrobial compounds are in use today, many have specific modes of action and, thus, have a narrow effective spectrum in terms of the bacterial species that they can target [13]. This issue can reduce their suitability in medical settings, which ideally require a more broad-spectrum treatment, given the frequent occurrence of co-infections [14]. Furthermore, selecting a narrow-effect spectrum that relies on a singular mechanism of microbial inactivation or inhibition can also make it easier for exposed bacteria to develop unwanted resistance. Thus, the use of two or more treatments in combination to treat bacterial infections represents a highly promising avenue of research. Checkerboard assays are well-documented methods used to assess the effects of different treatments when used in combination, whereby serially diluted concentrations of treatments are combined across a 96-well microtiter plate [15,16,17,18]. The resulting effects of combination therapy can be described as synergistic, additive, or antagonistic [16,19,20]. Synergy describes a total effect greater than the sum of the individual effects. An additive effect shows that the combined drugs exhibit a total effect equal to the sum of the individual effects, being no lesser or greater. An antagonistic effect describes combinations in which the total effect is lessened compared to the sum of the individual effects [21]. Combination therapies that result in an overall synergistic effect can allow a much greater impact to result from treatments that would normally hold less or, perhaps no, effect when used alone, such as in the case of AMR bacteria. While co-treatment therapies have been widely used in the treatment of diseases such as cancer, there is a rising interest in the synergistic abilities of previously established antimicrobial compounds [10,12,16]. 

In a previous study reported by these authors, the individual antibacterial capabilities of four GRAS bioactive compounds—silver nitrate (AgNO_3_), nisin, chitosan and zinc oxide (ZnO)—were assessed against a number of type-strain bacterial species, as well as AMR wild-type strains [22]. These GRAS bioactives were unusual in the sense that they withstood temperatures used to extrude and process polymers used in the manufacturing of medical devices; thus, these bioactives offer interesting options for new therapeutic research. In the present study, these four bioactives will be assessed regarding their antimicrobial capabilities in combination with one another, using arrangements of two-, three- and four-drug combinations. For this initial combinational study, three standard type strains will be used, namely *Escherichia coli*, *Staphylococcus aureus* and *Staphylococcus epidermidis*. *E. coli* and *S. aureus* were chosen as they represent Gram-negative and Gram-positive bacteria, respectively. *S. epidermidis* was included as it represents opportunistic Gram-positive bacterial pathogens and was observed to hold atypical behaviour against these four compounds in the previous study relative to *S. aureus*. The antimicrobial abilities of the two-drug combinations will be determined via use of a standard broth microdilution protocol in a checkerboard assay format, through which growth will be measured using turbidity absorbance readings. Three- and four-drug combinations will be assessed via use of novel versions of the checkerboard assay developed in the present study. The readings will be used to calculate the % growth of each treatment relative to the 100% growth control. While the checkerboard assay is a method commonly utilised to assess combination effects, there are various methods and programs developed for analysis of results [15,16,17,18]. The end results identified in the present study will analysed via use of the recently developed “synergy” python package, which can analyse large amounts of combinations and report their synergy scores [23].

## 2. Materials and Methods

### 2.1. Bioactive Solution Preparation

Silver nitrate (AgNO_3_) (SKU: S8157, CAS: 7761-88-8), nisin, 2.5% (SKU: N5764, CAS: 1414-45-5), chitosan of low molecular weight (SKU: 448869, CAS: 9012-76-4), zinc oxide (ZnO) and nanopowder of <100 nm in particle size (SKU: 544906, CAS: 1314-13-2) were purchased from Sigma-Aldrich/Merck (Merck Life Science Limited, Arklow, Co. Wicklow, Ireland). Chitosan was dissolved in 1% (*v*/*v*) acetic acid and adjusted to pH 5.5 using 0.4 M sodium hydroxide (NaOH). ZnO was suspended in dH_2_O. Nisin was dissolved in a solution of 400 mM sodium chloride (NaCl), which had a pH of 3.25. These solutions were then sterilised through autoclaving. Nisin concentrations were reported in terms of active nisin content, with 1 g of commercial nisin powder containing 25 mg of active nisin. AgNO_3_ was placed into a solution of 28% (*v*/*v*) Poly (ethylene glycol), which had an average molecular weight of 400 (PEG-400) and 26% (*w*/*v*) d-sorbitol [24]. This solution was then filter sterilised through use of a 0.2-micrometer syringe filter tip.

### 2.2. Bacterial Cell Culture

The standard strains, namely *E. coli* (ATCC 25922, NCTC 12241) and *S. aureus* (ATCC 29213, NCTC 12973), were purchased from Public Health England (Culture Collections, Public Health England, Salisbury, UK). *S. epidermidis* (ATCC 35984) was purchased from ATCC (LGC Standards, Middlesex, UK). Cultures were prepared via overnight incubation using tryptone soy agar (TSA). Colonies were then suspended in Mueller–Hinton broth (MHB) to 0.5 MacFarland absorbance for use as inoculum [25,26].

### 2.3. Two-Drug Combinational Broth Microdilution Assay

All steps were conducted under aseptic conditions or in closed systems. The antimicrobial properties of each bioactive solution in combinations of two were assessed in terms of their growth inhibitory capabilities, as determined via use of the broth microdilution method adapted from a previously published protocol [26]. Broth microdilution assays were carried out in flat bottom 96-well plates (untreated) against three chosen bacterial strains, namely *E. coli*, *S. aureus* and *S. epidermidis*. Before use, microplate lids were treated using a hydrophilic coating (20% (*v*/*v*) of isopropyl alcohol (IPA), 0.5% (*v*/*v*) of Triton-X100) [27]. Bacterial inoculums were prepared to give a final in-well concentration of 5 × 10^5^ cfu/mL, as determined via absorbance readings. Two-drug combination assays were prepared in an 8 × 8 checkerboard layout, allowing a total of 64 combinations. Dilutions of drug A and B were prepared in Mueller–Hinton broth (MHB) at a concentration four times higher (4×) than the highest desired final in-well concentration. Serial dilutions (1:2) of drug A and drug B were prepared in separate 96-well plates and combined in the final test plate (1:2 dilution) (See Appendix A). Each well was then inoculated with the prepared bacterial inoculum (1:2 dilution). Absorbance of the plate was measured using a BioTek^®^ Synergy HT microplate reader and Gen5 Microplate Reader Software (Version 2.01.14) (BioTek^®^ Instruments GmbH, Bad Friedrichshall, Germany). The plate was read using an endpoint absorbance read at 625 nM, and results were recorded as time-point 0 (t = 0) before incubation. This process allowed measurement of any turbidity caused by treatments and was be used as a blank. The plate was placed in a container to help prevent loss of well volume due to evaporation. The container was placed on a rotary incubator at 120 RPM, 37 °C for 18 h. Following incubation, plate absorbance was read (variable shake, and the 1-min endpoint absorbance was read at 625 nm). Results were recorded as timepoint 18 (t = 18). The absorbance values were used to calculate the % inhibition for each treatment well.

### 2.4. Three-Drug Combinational Broth Microdilution Assay

Three-drug combination assays were carried out as per the two-drug combination assay, albeit using a 6 × 6 checkerboard layout. Six such checkerboards were prepared by combining drug A and drug B, as per a two-drug combination assay, and different concentrations of drug C was added to each individual checkerboard. This setup allowed 6 concentrations of drug A, drug B and drug C to be assessed in combination (6 × 6 × 6), with a total of 216 combinations (See Appendix A for example layout). The experiment was split across three 96-well plates, allowing two 6 × 6 checkerboards per plate. A separate broth microdilution assay of single treatments was also carried out as a control to ensure that the treatments and bacteria tested performed in a nominal manner. Incubations and absorbance readings were carried out as per two-drug combinational assay.

### 2.5. Four-Drug Combinational Broth Microdilution Assay

Four-drug combination assays were carried out in a 4 × 4 checkerboard layout, which built on the three-drug layout design. The layout was designed in such a way that four 4 × 4 checkerboards (CBs) were set up within four 96-well plates. Each CB combined drug A with drug B. Each of these four CBs then had a different concentration of drug C added to it. To all CBs within each plate, a difference concentration of drug D will be added. The resulting system will yield a 4 × 4 × 4 × 4 combination (totalling in 256 combinations) (see Appendix A for example layout). 

A separate broth microdilution assay was also carried out using drugs A, B, C and D in tandem with the four-drug combination assay, which was used as a control to ensure that the individual treatments and bacteria being tested performed in a nominal manner. Plate and inoculum preparation, incubations and absorbance readings were all carried out as per the two-drug combinational assay.

### 2.6. Analysis of Results for the Determination of Synergy/Antagonism

Results from drug combination assays were analysed to determine drug interactions in terms of synergy or antagonism via the “synergy” python package [28]. Input data for synergy were prepared in an excel document using the concentration of each drug (µg/mL) and the % growth. Input data contained an individual column for the concentration of each drug (“drug1.conc”, “drug2.conc”, “drug3.conc” or “drug4.conc”). The Bliss model was chosen due to its simplicity and ability to analyse four-drug combinations. The reported response was expressed in terms of % growth. The response was input under the column “effect” and expressed as a decimal fraction of 1 (i.e., 100% growth = 1.0, 50% = 0.5, 0% = 0.0). Data were then exported as a .csv file. The synergy package was opened and run using PyCharm (version 2020.2) (JetBrains s.r.o, Prague, Czech Republic), which is a python-integrated development environment (IDE). Following the synergy documentation, input data were imported and analysed using the Bliss model. Results were expressed in terms of a synergy score, with a positive score indicating synergy, a score of 0 representing no effect and negative scores representing antagonism.

## 3. Results

Due to the number of combinations analysed during this study, only the three highest-scoring interactions of each combination and their average values will be reported and discussed. Synergy scores represent the magnitude of the combination interactions, where a positive score indicates synergy, scores close to 0 indicate an additive effect, a score of 0 represents no effect and negative scores represent antagonism. 2D heat-maps of all 64 combinations of each two-drug combination against each test bacterial species are presented in Figure 1, Figure 2, Figure 3, Figure 4, Figure 5 and Figure 6 showing each combination’s synergy score based on a colour scale, which is shown in the legend. Bar graphs have been prepared and presented in Figure 7, Figure 8 and Figure 9 for each two-drug, three-drug and four-drug combination respectively, showing the three highest-scoring drug combinations (Combo 1–3 on the *x*-axis) at the drug concentrations (µg/mL) shown on the left *y*-axis, with the calculated Bliss synergy score shown on the right *y*-axis (see Appendix A for the graphed data). 

### 3.1. Two-Drug Combinations

AgNO_3_–Chitosan

AgNO_3_ and Chitosan reported good synergistic interactions against each bacterial strain. The combination reported the highest average synergy scores against *E. coli* (average 0.4) and *S. aureus* (average 0.32). While the average concentration of chitosan was similar to that of the MIC versus *E. coli*, AgNO_3_ was reported to be present in lower concentrations. The most effective combination versus *S. aureus* reported concentrations that were 1/2 the MIC, with inhibition being approximately 69%. Results versus *S. epidermidis* reported good overall synergy, as much lower concentrations of each treatment exhibited more effective inhibition, and the second reported combination exhibited 99% inhibition, with 1/3 of the MIC of AgNO_3_ and less than 1/2 of the MIC of chitosan being used.

AgNO_3_–Nisin

AgNO_3_ and Nisin demonstrated a number of highly synergistic combinations (an average 0.32 versus *E. coli* and an average 0.24 versus *S. aureus*), as well as reporting the highest two-drug score from this study (average 0.68 versus *S. epidermidis*). While the highest-scoring combinations versus *E. coli* did not report inhibition exceeding 70%, there was moderate synergy observed compared to AgNO_3_ used alone at the same concentrations. The third highest-scoring combination versus *S. aureus* reported 99% inhibition, using less than 1/4 MIC of AgNO_3_ and 1/10 MIC of nisin. The three highest-scoring combinations versus *S. epidermidis* indicated that a concentration of 10 µg/mL AgNO_3_ was most effective in enabling nisin, which was reported to be present in relatively low concentrations, while still having a notable effect upon bacterial growth.

AgNO_3_–ZnO

AgNO_3_ and ZnO reported moderate synergy against *E. coli* (average 0.22) and *S. aureus* (average 0.26), as well as relatively high synergy versus *S. epidermidis* (average 0.44). The highest reported *E. coli* combination exhibited 98.5% growth inhibition at an AgNO_3_ concentration 1/4MIC and a ZnO concentration 1/2.5MIC, demonstrating a noticeable increase in the efficacy in both treatments. *S. aureus* results reported that lower concentrations of both AgNO_3_ and ZnO exhibited greater effect when combined. One reported combination exhibited 95.5% growth inhibition using 1/1.8MIC AgNO_3_ and 1/2.5MIC ZnO. AgNO_3_ and ZnO demonstrated the second highest-scoring average of all two-drug combinations (average 0.44) versus *S. epidermidis*. Reported combinations exhibited effective growth inhibition at much lower concentrations, even reaching 95.4% growth inhibition with 1/1.6MIC AgNO_3_ and 1/3.33MIC ZnO.

Nisin–Chitosan

Nisin and chitosan reported mixed results in combination. The highest-scoring combinations were identified versus *S. aureus* (average 0.24); however, the highest inhibition of these combinations reached only 50%, with no major reductions being seen in the concentrations of nisin or chitosan. Results versus *E. coli* show that greater concentrations of chitosan were needed to enable nisin; however, these concentrations exceeded the MIC of chitosan, making the combination ineffective. Results versus *S. epidermidis* demonstrated no major interactions, being close a synergy score of 0 in all combinations. Only one combination reported effective synergy, which exhibited 87.3% inhibition with a score of 0.11; however, the concentration of nisin used in this combination exceeded that of its MIC when tested alone.

Chitosan–ZnO

Chitosan and ZnO reported very few synergistic interactions versus *E. coli* (average 0.11), *S. aureus* (average 0.09) and *S. epidermidis* (average 0.27). Analysis of interactions versus *E. coli* show that a high concentration of chitosan was required for synergy to be identified; however, the amount of chitosan was 2 × MIC, and the synergy score was relatively low. While synergy was seen versus *S. aureus* at quite low concentrations of the two highest-scoring combinations, the exhibited growth inhibition was not noteworthy (3.1%, 2.6%, respectively). Combinations versus *S. epidermidis* reported moderate synergy at quite low concentrations of each combination; however, the inhibition did not exceed 31%.

Nisin–ZnO

Nisin and ZnO reported low synergy versus *E. coli* (average 0.08), *S. aureus* (average 0.06) and *S. epidermidis* (average 0.14). The highest-scoring combination versus *E. coli* (0.09) did not yield noteworthy inhibition, while the next highest-scoring combinations reported concentrations of ZnO that exceed the MIC in order to enable nisin. Highest-scoring combinations versus *S. aureus* reported low concentrations of each treatment; however, they had no noteworthy growth inhibitory effect (3.7–14%). Combinations versus *S. epidermidis* exhibited moderate inhibitory effects; however, the concentrations of ZnO exceeded that of the average MIC, and concentrations of nisin were not much lower that the previously reported MIC average.

### 3.2. Three-Drug Combinations

Chitosan–AgNO_3_–Nisin

Chitosan, AgNO_3_ and nisin reported moderate-to-high synergy in growth inhibition versus *E. coli* (average 0.38), *S. aureus* (average 0.56) and *S. epidermidis* (average 0.43). While the higher scoring combinations versus *E. coli* included high concentrations of chitosan (80–160 µg/mL), reported concentrations of AgNO_3_ were low (2–4 µg/mL) and had 99% inhibition. Concentrations of nisin were rather high, relative to those of the other test species (3.91–7.81 µg/mL). The highest-scoring combination versus *S. aureus* reported relatively low concentrations of each compound (78.13 µg/mL chitosan, 8 µg/mL AgNO_3_ and 10.63 µg/mL nisin) and expressed 99% inhibition. The second highest-scoring combination showed similar concentrations and levels of inhibition; however, when used twice, the amount of AgNO_3_ (16 µg/mL) was still less than the previously reported MIC. The third highest-scoring combination reported lower chitosan (39.06 µg/mL), though it did not fully inhibit *S. aureus* growth (71.45%). These combinations versus *S. aureus* reported the second highest average score of all three-drug test combinations. Combinations versus *S. epidermidis* reported near full inhibition (92–96%), as well as good synergy and low concentrations of chitosan (39.06–78.13 µg/mL) and nisin (0.63–1.25 µg/mL); however, concentrations of AgNO_3_ were near to the MIC (8–16 µg/mL). 

Chitosan–AgNO_3_–ZnO

Chitosan, AgNO_3_ and ZnO held moderately low synergy versus *E. coli* (average 0.14), *S. aureus* (average 0.28) and *S. epidermidis* (average 0.35), with combinations versus *E. coli* having the lowest scores of all three-drug combinations. Scores for *E. coli* were quite low, and each of the reported combinations exhibited full inhibition (97.15–100%) at low concentrations of chitosan (40–80 µg/mL), AgNO_3_ (0.5–4 µg/mL) and ZnO (20 µg/mL). The highest scores versus *S. aureus* reported consistently low concentrations of ZnO (62.5 µg/mL), as well as low concentrations of chitosan (39.06–78.13 µg/mL) and AgNO_3_ (8–16 µg/mL), while growth inhibition was high (97.55–98.32%). Combinations versus *S. epidermidis* reported good growth inhibition (87.95–98.4%) at relatively low concentrations of chitosan (20–40 µg/mL), AgNO_3_ (2–4 µg/mL) and ZnO (10 µg/mL). 

Nisin–AgNO_3_–ZnO

Nisin, AgNO_3_ and ZnO reported strong synergy versus *E. coli* (average 0.36), *S. aureus* (average 0.38) and *S. epidermidis* (average 0.53). Results versus *E. coli* show that very high concentrations of nisin (31.25 µg/mL) yielded high synergy at low concentrations of AgNO_3_ (0.5–1 µg/mL) and ZnO (31.25–62.5 µg/mL); however, growth inhibition did not exceed 77%. Combinations versus *S. aureus* reported moderate synergy at low concentrations of nisin (0.63–1.25 µg/mL) and moderate concentrations of AgNO_3_ (4–16 µg/mL) and ZnO (39.06 µg/mL). Combinations versus *S. epidermidis* reported the third highest average score synergy at low concentrations of nisin (1.25–2.5 µg/mL), AgNO_3_ (2–4 µg/mL) and ZnO (5 µg/mL); however, these combinations exhibited low-to-moderate growth inhibition (33.6–72.3%). 

Nisin–Chitosan–ZnO

Nisin, chitosan and ZnO reported low inhibition synergy versus *E. coli* (average 0.23) and *S. epidermidis* (average 0.21); however, combinations versus *S. aureus* reported the highest synergy score across all three-drug combinations (average 0.83). Concentrations of the reported combinations versus *E. coli* indicated poor synergy between treatments, as high concentrations of chitosan (9.77–312.5 µg/mL) and ZnO (31.25–125 µg/mL) were utilised. Combinations that used low concentrations of each drug exhibited very low inhibition (26.6%). 

The highest-scoring combination (0.93) versus *S. aureus* reported low concentrations of nisin (3.91 µg/mL), chitosan (39.06 µg/mL) and ZnO (62.5 µg/mL) with high inhibition (98.7%). The second highest-scoring combination (0.84) also reported low concentrations of nisin (0.977 µg/mL), chitosan (156.25 µg/mL) and ZnO (62.5 µg/mL) with high inhibition (99.4%). While the third highest-scoring combination reported low concentrations of nisin (0.977 µg/mL), chitosan (78.13 µg/mL) and ZnO (62.5 µg/mL), the reported growth inhibition was moderate (71.5%). Moreover, while results versus *S. epidermidis* reported low concentrations of nisin (0.98–1.95 µg/mL), chitosan (39.06 µg/mL) and ZnO (31.25 µg/mL), along with high growth inhibition (99%), there was little synergy observed, as denoted by the two highest scores. The third highest-scoring combination reported a very high concentration of chitosan (625 µg/mL).

### 3.3. Four-Drug Combinations

AgNO_3_–Nisin–Chitosan–ZnO

Chitosan, nisin, AgNO_3_ and ZnO exhibited moderately high synergy in combination versus *E. coli* (average 0.36) and very high synergy versus *S. aureus* (average 0.91) and *S. epidermidis* (average 1.11). While the compound’s average synergy score versus *E. coli* is lower than that versus the other two test species, results indicate the positive contributions of each treatment at low concentrations of chitosan (80 µg/mL), nisin (1.95–31.25 µg/mL), AgNO_3_ (8 µg/mL) and ZnO (10–40 µg/mL), which showed effective inhibition (69–98.9%). The highest-scoring combination, which exhibited 98.9% inhibition, reported a very high concentration of nisin (31.25 µg/mL), indicating that nisin had a strong influence within the combination. 

Reported synergy scores versus *S. aureus* are quite high, with the top scoring combinations exhibiting effective inhibition (97.5–99.5%) at low concentrations of chitosan (19.53–78.13 µg/mL), nisin (0.39–1.56 µg/mL) and AgNO_3_ (4 µg/mL) and moderate concentrations of ZnO (62.50 µg/mL). 

Combinations versus *S. epidermidis* reported the highest synergy scores identified within the present study at low concentrations of chitosan (20–80 µg/mL), nisin (1.25 µg/mL), AgNO_3_ (8 µg/mL) and ZnO (10–40 µg/mL). While the top combination reported a very high synergy score (1.3), the reported inhibition greatly deviated (stdev 69.18), having an average value of 24.96%. The second highest (1.13) and third highest (0.9) scoring combinations exhibited stable inhibition (98.7–99.1%) at similarly low concentrations of each treatment.

## 4. Discussion

Antimicrobial synergy holds great promise as a solution for use in meeting the AMR crisis for several reasons. While a bacterial species may hold or even develop resistance to a single therapeutic agent, co-treatment with an alternative compound that exhibits alternative modes of antimicrobial action could help to alleviate this issue. Additionally, certain groups of bacteria hold intrinsic metabolic or physical characteristics that can prevent certain classes of antimicrobials from exhibiting their effect. Co-treatment using a compound that can disrupt these characteristics would allow the primary treatment to carry out its effect unimpeded. Gram-negative bacteria are an example of one such group, as they have an additional outer membrane that can act to prevent compounds from reaching their target ligands. Following this example, nisin is a poly-cyclic lantibiotic that targets the inner-membrane-bound lipid II molecule. Due to the presence of an outer membrane, nisin is prevented from reaching its target, thus rendering it ineffective [29,30]. However, in theory, it would be possible to enable nisin by combining it via treatment with an additional compound that targets the outer membrane. By removing the outer membrane or compromising its integrity, nisin could freely interact with its lipid-II target. While this interaction can be clearly deemed to be synergistic, it is not enough on its own to observe a positive end result from the combination. While it would stand to reason that combining two or more already well-known and effective treatments would produce a greater gross effect than that of each individual treatment, previous studies of drug combinations have shown this predicted outcome to be incorrect, as have the results presented in this study [16,20,31]. To determine the synergistic abilities of two or more compounds, it is necessary to assess an array of various concentrations in different combinations. It is not important to determine the highest effect of combined treatments; the concept is to instead determine combinations that express a higher effect than that of the individual drugs at an identical concentration. The aim is to more easily discern the ratio of each drug required to enable another’s mechanism of action, thus giving the most efficient synergy. 

### 4.1. Inhibition and Synergy

Previously, AgNO_3_ was shown to be the most effective bacterial growth inhibitor of the tested bacterial species [22]. Nisin was shown to have very efficient inhibitory effects on test Gram-positive bacterial species, while having no effect on Gram-positive bacteria. Both compounds differ majorly in their modes of action, with AgNO_3_ permeating bacterial membranes through reactive silver ions (Ag^2+^), while nisin has specific binding affinity to the lipid-II molecules bound in the inner bacterial membrane. Nisin’s inability to affect Gram-negative bacteria is based on its inability to breach its outer membrane and interact with the lipid-II ligand. By combining both AgNO_3_ and nisin, it was hypothesised that the reactive Ag^2+^ ions of AgNO_3_ breach the Gram-negative outer bacterial membrane, allowing nisin to reach its target ligand [32,33,34,35]. Similar hypotheses were devised regarding ZnO and nisin, as ZnO had efficacy against Gram-negative and Gram-positive bacteria, as well as a similar mode of action wherein it destabilises membranes through release of Zn^2+^ ions and reactive oxygen species (ROS) [36,37,38,39]. Chitosan also had a noteworthy effect on all test strains, though it also had an alternate mechanism via which it targeted the bacterial cell wall [40,41]. The varying mechanisms had great significance for combinational studies and allowed us to observe whether effects unlock one other treatments’ drawbacks (AgNO_3_–nisin, ZnO–nisin), stack upon against them (AgNO_3_–ZnO) or complement them (chitosan–nisin, chitosan–AgNO_3_, chitosan–ZnO). 

In this study, the combinatory compatibility of four chosen bioactives was successfully established, as were the magnitude of their interactions with one another. The checkerboard assay was utilised to screen the inhibitory effect of bioactive combinations against each test bacterial strain. The checkerboard assay was a well-established method used to screen drug combinations in various areas of clinical research [12,15,16,17,18]. Through the use of the synergy python package and the Bliss model, the synergy score of each test combination was successfully determined. The results of this study presented interesting interactions between the bioactives, many of which were predictable, though others were unanticipated. 

### 4.2. Two-Drug Combination Synergy

The results of two-drug combination studies carried out against the Gram-negative bacteria *E. coli* have yielded varying results. Nisin, which is a lantibiotic that targets the inner-membrane-bound lipid II molecule, is hindered by the outer membrane found in Gram-negative bacteria, which prevents nisin from carrying out its mechanism of action. It was hypothesised that combining nisin with a compound capable of penetrating the outer membrane, such as AgNO_3_ or ZnO, would enable nisin, with the resulting interaction being marked as synergetic. Combinations of nisin–AgNO_3_ exhibited moderate-to-high synergy (average 0.32), showing a consistent concentration of AgNO_3_ (8.49 µg/mL) to be the most accommodating compound for varying concentrations of nisin. While the inhibition ranged between 64 and 68% for this combination, it shows that nisin was able to have an effect upon a previously unaffected target. In contrast, ZnO was not found to enable nisin; rather, it appeared that nisin was antagonizing ZnO, as the concentrations of ZnO in the most synergistic combinations were higher than that of its previously determined MIC. Combinations of nisin–chitosan also exhibited undesired results, with higher concentrations of chitosan being utilised to observe an inhibitory effect. While such results are unfavourable, they still present a promising observation, showing that nisin had an effect on Gram-negative bacteria. Combinations of AgNO_3_–chitosan exhibited a strong synergistic interaction, with effects being evident at lower concentrations of AgNO_3_, which would indicate chitosan’s ability to enable it. Chitosan has also shown to enable ZnO, which also exhibited lower concentrations; however, these combinations scored quite low, which reflects the fact that the concentration of chitosan was quite high. 

Two-drug combinations used to inhibit *S. aureus* and *S. epidermidis* growth presented some moderate-to-strong synergistic combinations; however, there was a pattern of ZnO not effectively combining with nisin or chitosan. AgNO_3_ demonstrates itself to be the most effect bioactive, enabling all other bioactives with which it is combined, reporting lower concentrations with higher inhibition responses. The highest overall scoring two-drug combination involved nisin–AgNO_3_ versus *S. epidermidis*. Chitosan also demonstrated notable synergy with most bioactives, though it only effectively combined with ZnO against *S. epidermidis*, as much lower concentrations of both gave a greater response; however, the inhibition response was weak. 

### 4.3. Three-Drug Combinations against Gram-Negative Bacteria

Increasing the combination number can further alter the effect exhibited by treatments, as is evident from three-drug combinations. Combinations that included nisin were shown to demonstrate high synergy versus *E. coli* with near full inhibition. Following the two-drug analysis, it was predictable that chitosan–AgNO_3_–nisin would effectively synergise, presenting the highest-scoring combination versus *E. coli*. Furthermore, the relatively high concentrations of nisin in this combination showed that it had an active effect on *E. coli*, as it can be presumed to be heavily involved in enabling resistance (i.e., a concentration close to 0 would indicate little-to-no input). A more unpredictable result was seen regarding combinations that involved ZnO, as two-drug combinations demonstrated ZnO to be a poor component in combination, though three-drug combinations showed opposing results. Nisin–ZnO was the lowest scoring combination versus *E. coli*; however, with the inclusion of AgNO_3_ or chitosan, these combinations were the second and third highest-scoring three-drug combinations against *E. coli*, respectively. Most interestingly, the nisin–AgNO_3_–ZnO combination highlights how little AgNO_3_ was reported in the higher scoring combinations, while relatively high concentrations of nisin were reported. This result again indicates nisin’s active role in the combination, whereas AgNO_3_ is at too low a concentration to have an inhibitory effect. This result could also demonstrate the ability of AgNO_3_ to enable the mechanism of nisin. While chitosan–AgNO_3_–ZnO reported low scoring combinations, the results seemed to be promising, with low concentrations of all three bioactives and nearly full inhibition reported. This result, once again, does not follow the patterns observed in two-drug combinations of the same bioactives.

### 4.4. Three-Drug Combinations against Gram-Positive Bacteria

Three-drug combinations versus *S. aureus* and *S. epidermidis* offered interesting points of comparison. The chitosan–AgNO_3_–nisin combination scored highly versus both bacterial strains; however, concentrations of AgNO_3_ were quite high at low concentrations of nisin. Scores from combinations that involved ZnO also proved to be quite unpredictable versus Gram-positive bacteria. Combinations of nisin–ZnO and chitosan–ZnO against *S. aureus* scored quite poorly; however, nisin–chitosan–ZnO reported the highest score of all three-drug combinations. In contrast, this combination had the second lowest score against *S. epidermidis*. The individual concentrations were quite low, and the reported synergy scores were also quite low. An interesting observation of this combination is that it was also predictable based on the two-drug combinations of nisin–chitosan, nisin–ZnO and chitosan–ZnO, which produced synergy scores that very closely averaged that of the nisin–chitosan–ZnO synergy score. Concentrations versus *S. aureus* indicate that ZnO enabled the effects of chitosan and AgNO_3_; however, concentrations versus *S. epidermidis* did not indicate that any single bioactive enabled another bioactive, highlighting the even distribution of activity between the three bioactives.

Nisin–AgNO_3_–ZnO demonstrated strong synergy versus both Gram-positive bacteria. While the reported synergy was particularly high against *S. epidermidis*, the reported inhibitory effects were quite low. Two-drug reports show that nisin–ZnO combinations interacted very poorly, which implies that the influence of AgNO_3_ caused the three-drug combinations to more favourably interact, which was also predictable given the synergy scores of nisin–AgNO_3_ and AgNO_3_–ZnO. 

### 4.5. Four-Drug Combinations

While a four-drug combination gave valuable insights into interactions between all four treatments at once, the 4 × 4 sized checkerboard had some disadvantages relative to larger 6 × 6 or 8 × 8 checkerboards, primarily the fact that it could not accommodate enough combinations to generate a full model of the possible combinational interactions. However, if determined via a two- or three-drug assay, key concentrations can be selected and utilised within the four-drug assay for further investigation. As such, the present four-drug assay layout should be used as a follow-on study, rather than as an initial combinational study, due to such limitations. Likewise, with the chosen three-drug assay layout, the four-drug assay layout allowed reduced experimental size and a faster setup.

Four-drug combinations reported a marked increase in the efficacy of all four bioactives relative to their individual capabilities against each bacterial strain. The combination of chitosan–nisin–AgNO_3_–ZnO against *E. coli* exhibited some predictable results, with synergy scores comparable to scores derived from two- and three-drug combinations. While concentrations of each bioactive in the highest-scoring combinations were lower than their individual MICs, concentrations of chitosan and AgNO_3_ were still quite moderate. Furthermore, only the highest-scoring combination reported complete inhibition while using a high concentration of nisin, which was expected due to nisin’s inability to target Gram-negative species.

The reported four-drug synergy scores against *S. aureus* and *S. epidermidis* were very high relative to other scores determined during this study. Concentrations versus *S. aureus* were notably lower than their individual MICs, thus having strong inhibitory effects. While *S. epidermidis* reported the highest synergy score of this study, its highest-scoring combination reported low inhibition. Compared to the other two reported combinations, a slight increase in either chitosan or ZnO was sufficient to push the effects toward complete inhibition, albeit remaining well below their individual MICs. From analysis of the most effective combinations against each individual bacterial strain, the most effective concentrations of each drug in combination were determined, which gave a four-drug combination that could cause complete inhibition against all tested bacterial strains, while the amount of each drug used was limited (See Table 1). 

## 5. Conclusions

Combinational antimicrobial bioactive studies hold great potential in many areas of clinical, pharmaceutical and medical research. The discovery of cross-treatment synergism could potentially unlock many new avenues of therapy for various pathogens and conditions. While drug synergy is a key target of combinational studies, drug antagonism is also a well-documented occurrence in pharmaceuticals, and while there are several models under development for its prediction, in many cases, it is difficult to determine which treatments may interact negatively without performing pre-clinical or clinical studies. Though it is important to find compatible combinations of drugs, another key goal is finding combinations in which the individual drugs are more effective within a combination than they are when acting alone. Determining synergy scores is an efficient method of screening many combinations of treatments and deducing the most effective option. It is evident from results presented here that treatment interactions cannot be accurately predicted and can differ greatly between bacterial strains. Furthermore, increasing the combination number has also been shown to have an unpredictable effect, as two-drug combinations cannot predict the effects of three-drug combinations of the same components, and likewise, two- and three-drug combinations cannot predict the effects of four-drug combinations. Current findings show that models previously used to predict drug combinations cannot be wholly trusted, as there are aberrant results presented in this study that contradict predictive models. While such results may not be considered advantageous, they provide knowledge critical to the development of combinational treatments. Within the scope of investigating new or previously unsuitable compounds as alternative antimicrobial treatments, the possibility of using as little of each compound possible while still holding an antimicrobial effect holds great potential. Identification of antimicrobial compounds based on complementary modes of action also provides an additional avenue for the discovery of novel treatments that combat AMR bacteria, wherein the activity of one treatment may enable the activity of another treatment that is naturally ineffective against the bacteria, as was the case of nisin against Gram-negative species in the present study.

There is also a pressing need to use effective screening methods to evaluate alternative bioactives that are biorefined via various environmental and food waste streams to help address the shortage in appropriate antimicrobials and the development of AMR [42,43]. Interestingly, there is increased interest in exploring new sources of antimicrobials, such as marine, peatland and food waste streams, that may present stressful environments that favor the production of unique antimicrobial bioactives [44]. This simple mass-throughput screening-based approach to evaluating combinational bioactives will also help address the surge in resistance to anti-fungal drugs among problematical fungi that cause significant human and animal infections [45,46]. There is also a proportionate interest in progressing interdisciplinary research through Quadruple Helix Hub frameworks (combining academia, industry, society and regulators), which use shared access to specialist equipment and subject-matter experts across disciplines to overcome these challenges [47].

The four chosen bioactives, namely AgNO_3_, ZnO, nisin and chitosan, which have previously been characterised in terms of their individual antimicrobial abilities, have now been characterised for their combinational interactions in two-, three-, and four-drug arrangements. Using this data, we accurately determined the most effective concentrations of each compound required for most effective microbial growth inhibition, limiting the amount of each compound needed to inhibit bacterial growth while having a broader spectrum of effect. Using this data, it is clear that the use of these compounds in combination produces a much more effective inhibitor of microbial growth, as the concentration of each individual bioactive is much lower than that of their MIC alone. As the current study relied on combinations of serial dilutions, more extensive analyses must be carried out regarding the most synergistic combinations, with further testing used to determine the most ideal concentration combinations and verify their efficacy. Additionally, the methods and analysis procedures presented have been shown to produce detailed and high-throughput assessments of drug combinations, and as such, they should be carried forward to evaluate additional treatments and combinations. The methodology could also be adjusted accordingly to allow studies of pharmaceuticals in other fields of research.

## Figures and Tables

**Figure 1 biomedicines-11-02216-f001:**
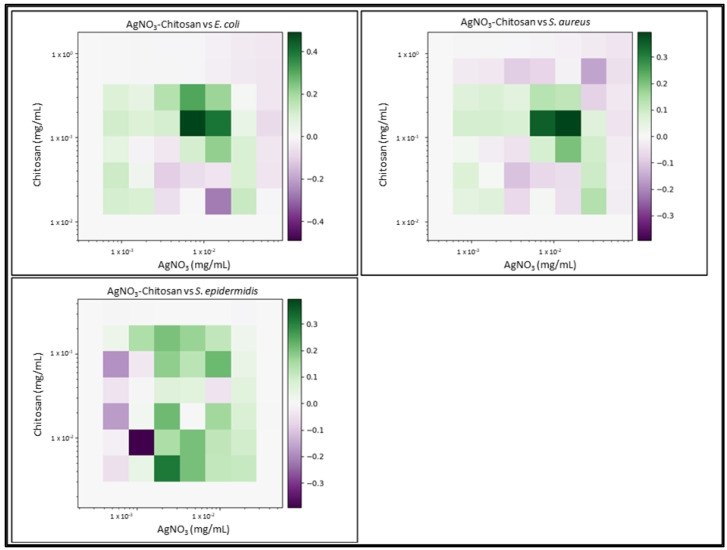
AgNO_3_–chitosan synergy heat map: Graphs show heat map of synergy between silver nitrate (AgNO_3_) and Chitosan in inhibiting *E. coli, S. aureus* and *S. epidermidis* growth as determined via broth microdilution and absorbance readings. Inhibition results were analysed using the synergy python package using the Bliss synergy model. The synergy python package produced the heatmap graphs of each combination result, giving visual presentations of combinations of synergy (green) or antagonism (purple). *n* = 3.

**Figure 2 biomedicines-11-02216-f002:**
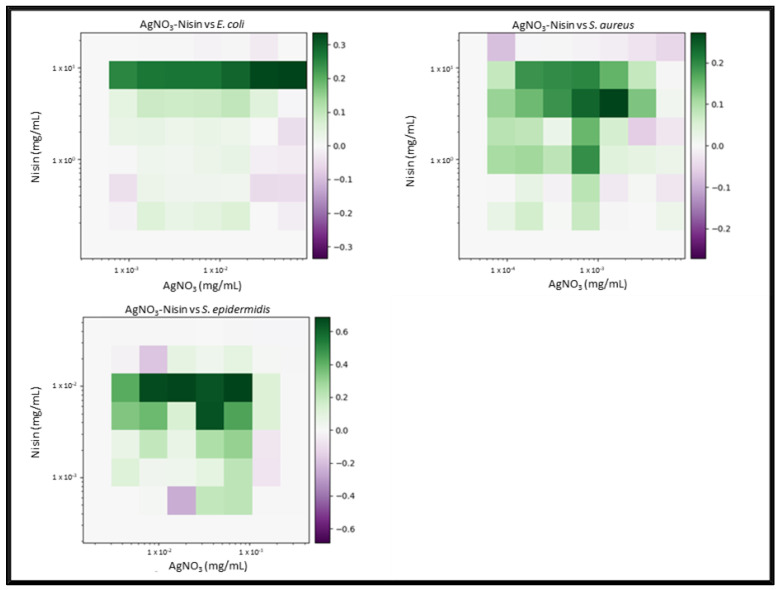
AgNO_3_–nisin synergy heat map: Graphs show heat map of synergy between silver nitrate (AgNO_3_) and Nisin in inhibiting *E. coli*, *S. aureus* and *S. epidermidis* growth as determined via broth microdilution and absorbance readings. Inhibition results were analysed with the synergy python package using the Bliss synergy model. The synergy python package produced the heatmap graphs of each combination result, giving visual presentations of combinations of high (green) and low (purple) synergy. *n* = 3.

**Figure 3 biomedicines-11-02216-f003:**
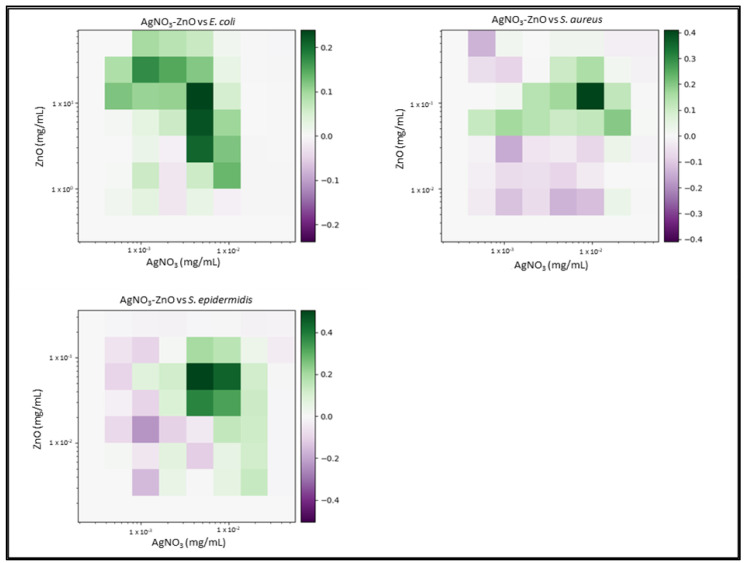
AgNO_3_–ZnO synergy heat map: Graphs show heat map of synergy between silver nitrate (AgNO_3_) and zinc oxide (ZnO) in inhibiting *E. coli*, *S. aureus* and *S. epidermidis* growth, as determined via broth microdilution and absorbance readings. Inhibition results were analysed via the synergy python package using the Bliss synergy model. The synergy python package produced the heatmap graphs of each combination result, giving visual presentations of combinations of high (green) and low (purple) synergy. *n* = 3.

**Figure 4 biomedicines-11-02216-f004:**
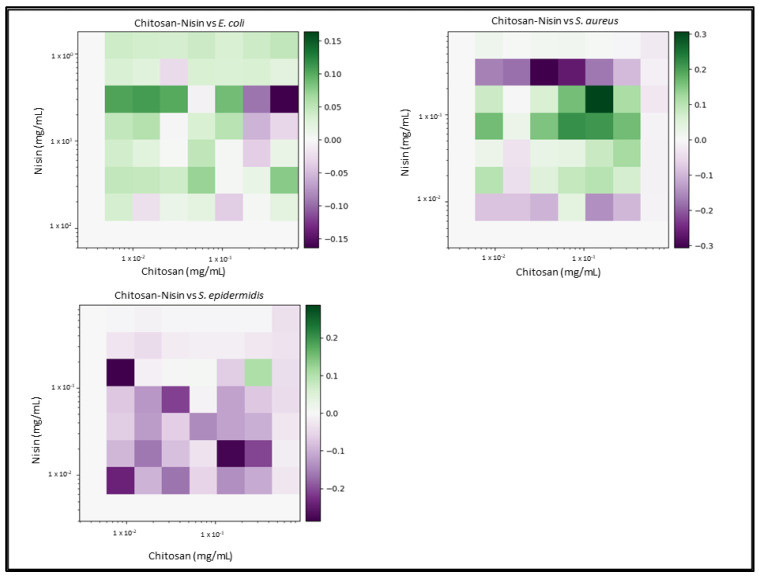
Chitosan–nisin synergy heat map: Graphs show heat map of synergy between Chitosan and Nisin in inhibiting *E. coli*, *S. aureus* and *S. epidermidis* growth, as determined via broth microdilution and absorbance readings. Inhibition results were analysed via the synergy python package using the Bliss synergy model. The synergy python package produced the heatmap graphs of each combination result, giving visual presentations of combinations of high (green) and low (purple) synergy. *n* = 3.

**Figure 5 biomedicines-11-02216-f005:**
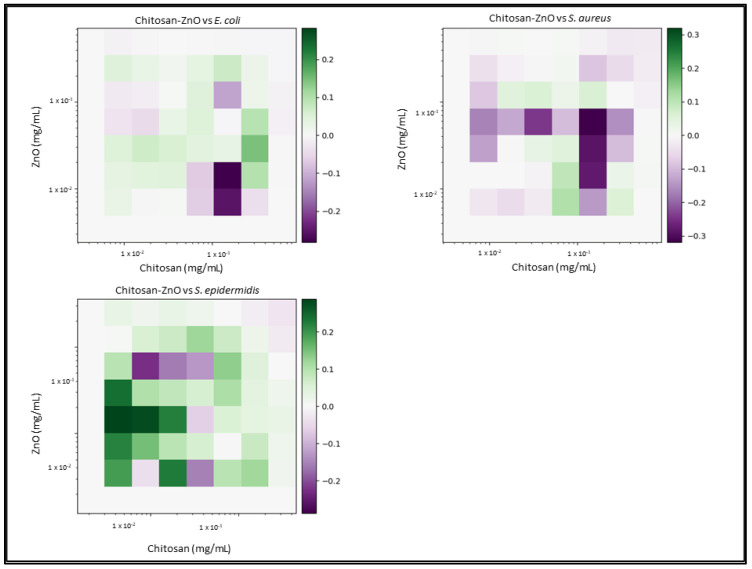
Chitosan–ZnO synergy heat map: Graphs show heat map of synergy between Chitosan and zinc oxide (ZnO) in inhibiting *E. coli*, *S. aureus* and *S. epidermidis* growth, as determined via broth microdilution and absorbance readings. Inhibition results were analysed via the synergy python package using the Bliss synergy model. The synergy python package produced the heatmap graphs of each combination result, giving visual presentations of combinations of high (green) and low (purple) synergy. *n* = 3.

**Figure 6 biomedicines-11-02216-f006:**
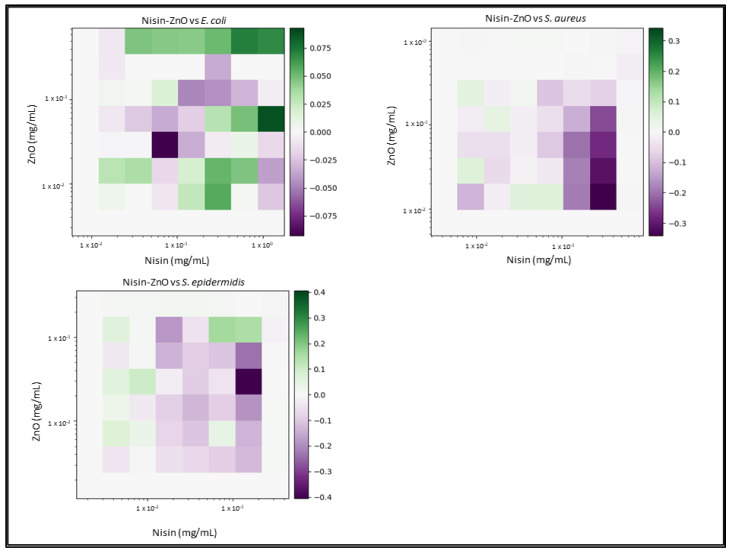
Nisin–ZnO synergy heat map: Graphs that show heat map of synergy between nisin and zinc oxide (ZnO) regarding the inhibition of *E. coli*, *S. aureus* and *S. epidermidis* growth, as determined via broth microdilution and absorbance readings. Inhibition results were analysed via the synergy python package using the Bliss synergy model. The synergy python package produced the heatmap graphs of each combination results, giving visual presentations of combinations of high (green) and low (purple) synergy. *n* = 3.

**Figure 7 biomedicines-11-02216-f007:**
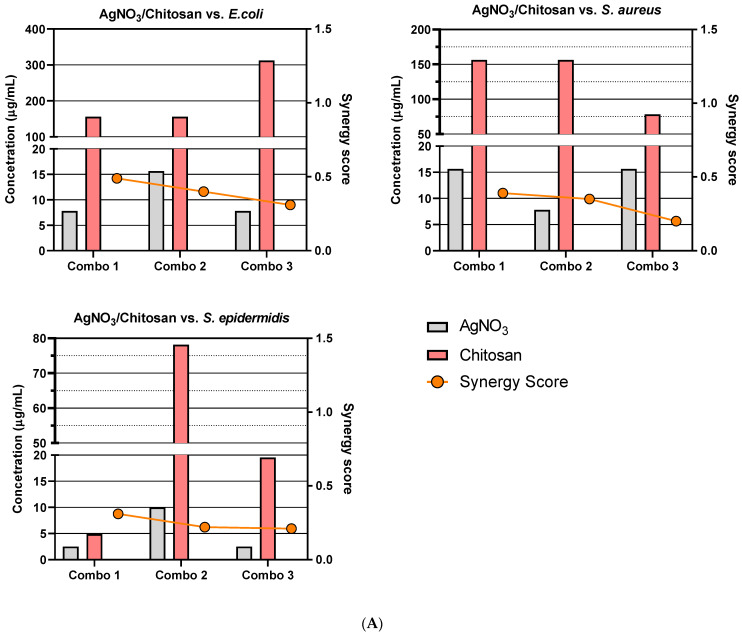
Synergy score results of the top three two-drug combinations: Bar graphs that present the concentrations of each drug and their respective synergy scores from the top three two-drug combinations (**A**–**F**) against *E. coli*, *S. aureus* and *S. epidermidis*. (**A**) AgNO_3_/Chitosan, (**B**) Nisin/AgNO_3_, (**C**) AgNO_3_/ZnO, (**D**) Nisin/Chitosan, (**E**) Chitosan/ZnO and (**F**) Nisin/ZnO. Bars show drug concentrations, as indicated on the left *y*-axis, and the line/symbols show each combination’s (combo) synergy score, as indicated on the right *y*-axis.

**Figure 8 biomedicines-11-02216-f008:**
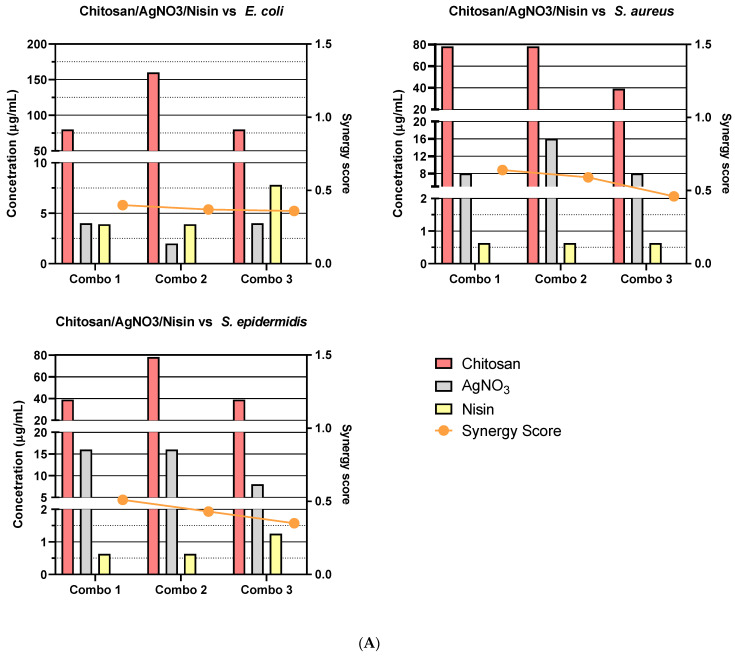
Synergy score results of top three three-drug combinations: Bar graphs presenting the concentrations of each drug and their respective synergy scores from the top three three-drug combinations (**A**–**D**) against *E. coli*, *S. aureus* and *S. epidermidis*. (**A**) Chitosan/AgNO_3_/Nisin, (**B**) Chitosan/AgNO_3_/ZnO, (**C**) Nisin/AgNO_3_/ZnO and (**D**) Nisin/Chitosan/ZnO. Bars show drug concentrations as indicated on the left *y*-axis and the line/symbols show each combination (combo) synergy score as indicated on the right *y*-axis.

**Figure 9 biomedicines-11-02216-f009:**
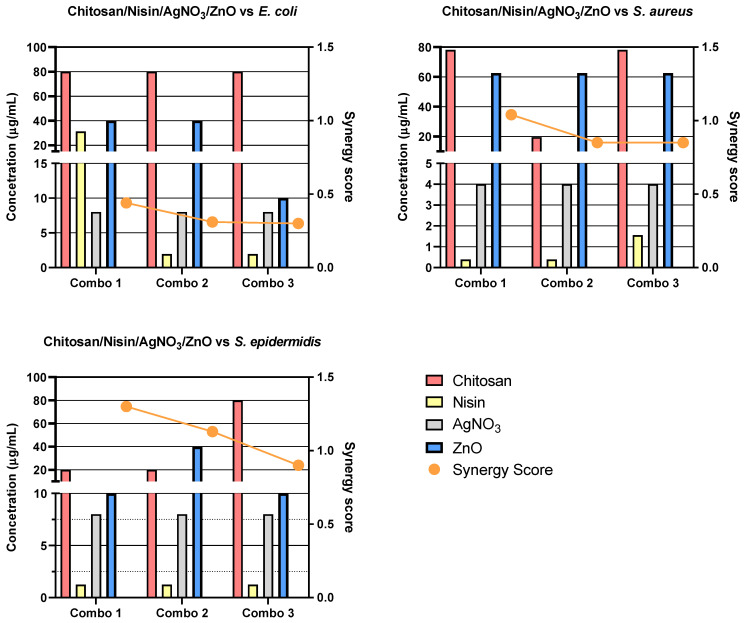
Synergy score results of the top three four-drug combinations: Bar graphs that present the concentrations of each drug and their respective synergy scores from the top three four-drug combinations (Chitosan/Nisin/AgNO_3_/ZnO) against *E. coli*, *S. aureus* and *S. epidermidis*. Bars show drug concentrations, as indicated on the left *y*-axis, and the line/symbols show each combination (combo) synergy score, as indicated on the right *y*-axis.

**Table 1 biomedicines-11-02216-t001:** Most effective concentrations of the four bioactive compounds in combination against *E. coli*, *S. aureus* and *S. epidermidis*. These concentrations were established by evaluating the highest-scoring combinations against each bacterial species and determining the lowest concentrations of each compound that would cause complete inhibition of all three bacterial species.

Bioactive	Most Effective Concentration (µg/mL)
Chitosan	80
Nisin	2
AgNO_3_	8
ZnO	60

## Data Availability

Not applicable.

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
