# Peer review of "Synergy Assessment of Four Antimicrobial Bioactive Compounds for the Combinational Treatment of Bacterial Pathogens"

_biomedicines, 2023, doi:10.3390/biomedicines11082216_

Round 1
Reviewer 1 Report
The manuscript entitled ‘Synergy assessment of four bioactive compounds for combinational treatment of antimicrobial resistant bacterial pathogens’ intends to assess the antimicrobial capabilities of the four bioactive agents (silver nitrate, nisin, chitosan, and zinc oxide) in combinations with one another, in arrangements of two-drug, three-drug and four-drug combinations using a checkboard assay. However, the manuscript requires major revisions prior to any possible consideration of this manuscript to be published in the journal of ‘biomedicines’.
1. Why have the authors chosen checkboard assay in this study? Since the checkboard assay is the primary method used for the synergy measurements of antimicrobics in this work, a brief description of it should be provided in the #Introduction section.
2. #Figure_1 in the main text is not enhancing the quality of the manuscript. The authors may put this figure in the Supplementary materials.
3. The antibacterial results (optimum/highest one) should be presented as bar diagram for better representation (important information at one place) of the findings.
4. The combination showing maximum antibacterial activity should be tested on bacterial plates (Mueller-Hinton Agar) and the pictures should be presented to evidence the findings of this study.
5. What do the authors mean by the sentence in #Line_586-588 of the ‘Conclusion’ section? The authors should avoid the general statements in the ‘Conclusion’ section and must highlight the important findings of this work.
6. As mentioned in #Line_592-594, “From this data, we can accurately determine the most effective concentrations of each compound for most effective microbial growth inhibition, limiting the amount of each needed to inhibit bacterial growth, while also holding a broader spectrum of effect.” The most effective concentrations of each sample tested in this study should be provided in a separate table.
7. Author suggested to cite some relevant papers and discus the combinational effect. For example,
https://doi.org/10.1038/s41598-022-14117-w, https://doi.org/10.1016/j.ijbiomac.2023.124129
8. The reference citation style should be consistent throughout the manuscript.
Minor editing of English language required
Reviewer 2 Report
Manterson and his colleagues conducted several experiments to assess the effect of combining various antimicrobial agents on different strains of bacteria.
Although the title refers to "treatment of antimicrobial-resistant bacterial pathogens", the authors do not provide information on the antibiotic susceptibility profile of the bacteria tested. It is not stated whether the strains of E. coli, S. aureus and S. epidermidis are clinical strains, ATCC, or of other origin. Is the strain of S. aureus MSSA or MRSA?
Without this indication, and if they are not strains with an established antimicrobial resistance profile, the title and conclusions are not supported by the results obtained by the authors.
It would also be interesting if the authors could explain why these three species were chosen.
In the material and methods section it is not stated where the drugs used and the culture media were purchased.
Minor
The name of the bacteria should be placed in italics throughout the text.
Round 2
Reviewer 1 Report
-
Reviewer 2 Report
The document has been revised and improved by the authors.